# A Household-Based Survey to Understand Factors Influencing Awareness, Attitudes and Knowledge towards *Wolbachia-Aedes* Technology

**DOI:** 10.3390/ijerph182211997

**Published:** 2021-11-15

**Authors:** Li Ting Soh, Zoe Ong, Kathryn Vasquez, Irene Chen, Xiaoxi Li, Weixin Niah, Chitra Panchapakesan, Anita Sheldenkar, Shuzhen Sim, Lee Ching Ng, May O. Lwin

**Affiliations:** 1Environmental Health Institute, National Environment Agency, 11 Biopolis Way, #06-05-08, Singapore 138667, Singapore; soh_li_ting@nea.gov.sg (L.T.S.); kathryn_vasquez@nea.gov.sg (K.V.); irenechen88@live.com.sg (I.C.); li_xiaoxi@nea.gov.sg (X.L.); niah_weixin@nea.gov.sg (W.N.); sim_shuzhen@nea.gov.sg (S.S.); 2Wee Kim Wee School of Communication and Information, Nanyang Technological University, 31 Nanyang Link, Singapore 637718, Singapore; ongz0063@e.ntu.edu.sg (Z.O.); chitrapk@ihpc.a-star.edu.sg (C.P.); anitas@ntu.edu.sg (A.S.); tmaylwin@ntu.edu.sg (M.O.L.); 3Global Asia, Interdisciplinary Graduate Programme, Nanyang Technological University, 61 Nanyang Drive, Academic Block North, ABN-01b-11, Singapore 637335, Singapore; 4Affective Computing Group, Institute of High Performance Computing (IHPC), Agency for Science, Technology and Research (A*STAR), 1 Fusionopolis Way, #16-16 Connexis, Singapore 138632, Singapore; 5School of Biological Sciences, Nanyang Technological University, 60 Nanyang Drive, Singapore 637551, Singapore

**Keywords:** *Wolbachia*, *Wolbachia-Aedes*, Project *Wolbachia*–Singapore, household survey, perception, dengue

## Abstract

In 2016, Singapore introduced the release of male *Wolbachia-Aedes* mosquitoes to complement vector control efforts and suppress *Aedes aegypti* mosquitoes in selected study sites. With ongoing expansion of Project *Wolbachia–*Singapore to cover larger areas, a household-based survey was conducted between July 2019 to February 2020 in two Project *Wolbachia* study sites using a structured questionnaire, to evaluate current sentiments and assess the need for enhanced public messaging and engagement. The association of factors that influence awareness, attitudes, and knowledge towards the use of *Wolbachia-Aedes* technology was analysed using Pearson’s Chi-square test and binary logistic regression. Of 500 respondents, 74.8% were aware of Project *Wolbachia–*Singapore. Comparatively, the level of knowledge on *Wolbachia-Aedes* technology was lower, suggesting knowledge gaps that require enhanced communication and messaging to address misinformation. Longer exposure to the project predicted greater awareness, whereas higher education levels predicted higher knowledge levels. Younger age groups and higher education levels were associated with high acceptance towards the project. High levels of trust and acceptance towards the project were also observed across the population. The public’s positive perception of the project is a testament to the effective public communication undertaken to date and will facilitate programme expansion.

## 1. Background

Dengue has rapidly spread in all World Health Organization (WHO) regions in recent years [1]. One study estimates the global burden of dengue to be 390 million dengue virus infection per year, of which 96 million (67–136 million) shows visible symptoms [2]. Asia bears about 70% of the global burden of dengue and continues to be plagued by large outbreaks. Located in Southeast Asia, Singapore is also not spared from the disease [3,4]. The country is a highly urbanised environment with a tropical climate that favours the proliferation of *Aedes aegypti* mosquitoes and transmission of the dengue virus [5], making Singapore vulnerable to dengue outbreaks. In 2020, Singapore experienced the largest dengue outbreak on record, with 35,315 dengue fever and dengue haemorrhagic fever cases and 32 dengue deaths [6]. Multiple factors contribute to the continued susceptibility of Singapore’s population to explosive dengue outbreaks, including reduced herd immunity after decades of low incidences [7], and the presence of cryptic or difficult to find *Aedes* mosquito breeding sites [8,9]. The limitations of the only commercially available vaccine Dengvaxia [1] has left vector control as the key strategy for epidemic control. There is an urgent need for novel and sustainable dengue control approaches to bring down the *Aedes aegypti* mosquito population, which is the primary vector for dengue in Singapore.

One such vector control approach is the incompatible insect technique (IIT) [10,11], which has been piloted in several countries such as Thailand (Chachoengsao), USA (Fresno), Australia (Innisfail), and Singapore [12,13,14,15]. This approach involves the release of male *Wolbachia-*carrying *Aedes aegypti* mosquitoes (*Wolbachia-Aedes*) into the environment. Mating between male *Wolbachia-Aedes* mosquitoes and female urban *Aedes aegypti* mosquitoes results in non-viable eggs, due to a phenomenon known as cytoplasmic incompatibility [16]. Continued releases of male *Wolbachia-Aedes* are expected to suppress *Aedes aegypti* mosquito population to a level that cannot sustain dengue transmission. In Singapore, the National Environment Agency (NEA), a statutory board under the Singapore Government, is conducting a multi-phased field study, termed “Project *Wolbachia–*Singapore”, to evaluate the efficacy of the technology and develop deployment strategies for high-rise housing estates [15,17]. Field releases began in 2016, with gradual expansion from 39 housing blocks to 144 housing blocks in 2019. The IIT approach is aligned with Singapore’s emphasis on source reduction, and is intended to complement traditional vector control measures, such as mosquito breeding habitat removal, space-spraying with insecticides, and vector surveillance [18,19,20]. The project has reduced the *Aedes aegypti* mosquito population by more than 98% and has reduced dengue cases by up to 88% in study sites with at least one year of releases [15].

Given the long-standing public advocacy on the importance of keeping mosquito populations low [21,22], the idea of releasing male *Wolbachia-Aedes* mosquitoes into the community may sound counterintuitive. Therefore, community engagement to educate the public on the project’s objectives and goals is an integral component that cannot be overlooked [18].

As such, in addition to rigorous laboratory studies and risk assessment, NEA continues extensive groundwork to share information on the project, as well as engaging and consulting with stakeholders. This included residents at the study sites, the general public, the medical and scientific community, and government agencies [18,23]. Adopting this consultative approach allows for the gathering of suggestions, feedback and concerns that forms the basis for communication strategies. Examples of engagement on the ground include sharing sessions, roadshows and exhibition booths at community areas with high footfalls, and mosquito production facility tours, to educate and garner support from residents. Information brochures are distributed to all households at the study sites, and publicity materials such as posters and banners are displayed at prominent areas to raise awareness of the project. These publicity materials are produced in the country’s four official languages—English, Chinese, Malay and Tamil. In addition, regular updates of the project are published through mainstream media and also on NEA’s webpage, and shared at community leaders’ meetings.

The benefit of *Wolbachia-Aedes* mosquito releases in public spaces extend to all residents within the study site, and to some extent the adjacent areas as well, regardless of the demography. However, the scale of deployment, infrastructure, environmental conditions and mosquito behaviour make it unfeasible to offer residents a choice of exemption from the study. This is unlike other studies, where releases of *Wolbachia-Aedes* mosquitoes are only carried out in sites with consent from all households in the community [24,25]. Therefore, several surveys were conducted in 2016 prior to and at the initial phase of the field trial to understand the level of public acceptance towards the project [23]. These surveys revealed that majority of the population had no objection to the releases of male *Wolbachia-Aedes* mosquitoes in their neighbourhoods. A household survey conducted during the first phase of the field trial in November to December 2016, showed that more than 70% of the study-site households interviewed had heard of the project, and more than 90% had no concerns with the releases [18,23]. Since then, the project has expanded more than 3 times the size of original study sites—from 39 housing blocks with approximately 3941 households in 2016, to 144 housing blocks with approximately 13,510 households in 2019. During this period, a transparent and responsive feedback management system was adopted to maintain trust and accountability to community stakeholders. Some feedback received included increased sighting of mosquitoes due to male *Wolbachia-Aedes* releases and nuisance caused by increased number of mosquitoes. Inconveniences such as having to close doors and windows to reduce an influx of mosquitoes, and additional spraying of insecticides and repellent to reduce swarming of male *Wolbachia-Aedes* indoors were also responses that were received. Therefore, it is essential to re-evaluate the sentiments on the ground at the approximate five-year mark and to assess the need for enhanced public messaging and engagement. In this study, a follow-up household perception survey was conducted to explore the awareness, knowledge and attitudes of the public towards Project *Wolbachia*–Singapore.

## 2. Methods

### 2.1. Study Area and Survey Collection Strategy

The household perception surveys were administered from July 2019 to February 2020, at two Project *Wolbachia*–Singapore study sites in Tampines and Yishun, where releases of male *Wolbachia-Aedes* mosquitoes are being carried out. Tampines and Yishun are located at the eastern and northern parts of Singapore respectively (Figure 1). The housing type and environment in both areas are similar, with apartments and commercial units developed by the Housing Development Board of Singapore.

The study areas are in the same locality as the earlier household survey conducted in the initial phase of the field trial in 2016 [23], but expanded to cover a total of 144 public housing apartment blocks in a highly urban setting (with more than 13,000 households in the areas). The earlier household survey covered 39 blocks (estimated 3941 households).

Within each site, the households are further classified into 2 categories—(i) households that have experienced >2 years of *Wolbachia-Aedes* releases; (ii) households that have experienced <1 year of *Wolbachia-Aedes* releases at the point of survey (Table 1). The minimum sample size for each category was 94 (95% CI, 10% margin of error), calculated using the Qualtrics® online sample size calculator [26]. To increase the representativeness and accuracy of our data, we increased the number of households surveyed per site proportionally, resulting in a total of 500 households surveyed. Surveys were conducted by randomly selecting a representative number of households from each block, based on a randomised household list generated. One respondent from each household was surveyed after obtaining his/her informed consent to participate. The next household on the randomised list was chosen if there were no respondents present at a particular household, or if the respondent was unwilling or unable to participate due to language barriers. Surveys were administered after obtaining the consent from the participant.

### 2.2. Data Collection

A total of 500 surveys were administered using a structured questionnaire, through face-to-face interviews. The closed-ended questionnaire consists of demographic characteristics, knowledge, attitudes and perceptions towards the project and the technology. All surveyors were trained prior to conducting the surveys, to ensure that the objectives, methodology, expectations and questions were well understood.

### 2.3. Measures

The levels of knowledge on *Wolbachia-Aedes* technology among participants was evaluated based on a series of 11 questions, and the “general knowledge (GK)” score was aggregated by adding up the number of correct responses. The GK scores were categorised into “Low” (0–3 points), “Mid” (4–7 points) and “High” (8–11 points) categories. Participants were also asked a series of questions to evaluate how well they perceived their knowledge of the topic, and the perceived knowledge score was obtained by adding up the number of times each participant replied “Yes” to the 6 questions under “perceived knowledge (PK)”. The PK scores were similarly categorised into “Low” (0–2 points), “Mid” (3–4 points) and “High” (5–6 points) categories.

Participants’ trust in the project was measured using 3 items on a 5-point Likert scale (1 = strongly disagree, 5 = strongly agree). Internal consistency of items under the Trust construct are acceptable (α = 0.850), and hence averaged to create the composite measure–Trust in project.

Participants’ acceptance towards Project *Wolbachia*–Singapore was assessed using a single item that asked how long participants were able to accept more mosquitoes in their environment in support of Project *Wolbachia*. Participants were given 5 options, ranging from one month to more than a year. Participants were also asked to indicate their gender, age group and highest education levels, as well as general perception of dengue as a problem in Singapore, their confidence in NEA, and whether they felt that NEA acts in the interest of residents.

### 2.4. Statistical Analysis

The relationships between GK and PK scores, along with demographic factors (gender, age, education levels and length of exposure to Project *Wolbachia*) and the outcome variables (greater awareness, high levels of general knowledge, trust and acceptance) were analysed using IBM SPSS Statistics (Version 27, Armonk, NY, USA). Pearson’s correlation (for total GK and PK scores) and Pearson’s Chi-square test (for categorised GK and PK scores) were conducted to compare GK and PK scores. Bivariate analysis was first conducted using Pearson’s Chi-square test to determine if any associations between the independent variables and outcome variables were significant. Multivariable logistic regression was then used to analyse the impact of relevant demographic factors on each outcome. As suggested by Alyousefi et al. [27], only independent variables that showed an association of *p* < 0.20 in Pearson’s Chi-square test were included for the multivariable logistic regression.

## 3. Results

The characteristics of the participants in the study are summarised in Table 2. There was an equal proportion of participants who had been exposed to the project for less than a year versus more than two years, but slightly more participants from the larger study site in Yishun (55%) compared to the study site in Tampines (45%) (Table 1).

### 3.1. Awareness of Project Wolbachia–Singapore

A total of 74.8% (n = 374) of the participants had heard of Project *Wolbachia–*Singapore, with the majority becoming aware of the project through publicity pamphlets distributed to all households, posters, and banners within study sites (46.8%), mainstream media such as TV news/radio (30.2%) and active engagement including roadshows and door to door engagement sessions (13.4%).

### 3.2. Knowledge on Project Wolbachia–Singapore

Among all the participants, 38.4% of them felt confident of their knowledge on the project with “High” “perceived knowledge” (PK) scores. The remaining fell into the “Mid” (20.6%) and “Low” (41.0%) categories. When asked on the facts related to *Wolbachia* technology and the project, only 17.2% of the participants were able to attain “High” “General Knowledge” (GK) scores, the remining scored in the “Mid” (40.2%) and “Low” (42.6%) categories.

Participants who were aware of Project *Wolbachia*–Singapore had a higher PK score (mean = 4.1, SD = 1.8) and obtained higher GK scores (mean = 5.2, SD = 2.6) when compared to participants who were not aware of the project (mean “PK” score = 0.4, SD = 1.0; mean “GK” Score = 1.8, SD = 2.0). The score distribution is summarised in Figure 2. We further evaluated the relationship between general knowledge and perceived knowledge on the topic of *Wolbachia* technology among participants who had heard of Project *Wolbachia*–Singapore, to avoid bias in comparison among those who were unaware of the project.

The majority (76.5%) of participants who had heard of the project were confident of their knowledge, scoring themselves between “Mid” to “High” levels in terms of their perceived knowledge. In terms of the level of general knowledge on the topic, 71.4% of these participants who had heard of the project scored (GK scores) between “Mid” to “High” levels. The performance of the participants in the PK and GK questions are summarised in Table 3. There is a moderate correlation (r = 0.512, *p* < 0.01) between the mean PK and GK scores. The relationship between PK and GK was further tested using Pearson’s chi-squared test on the categorised PK and GK scores, and they were found to be significantly correlated (*X*^2^(4, 374) = 74.41, *p* < 0.001). The majority of those with low perceived knowledge (59.1%) had a correspondingly low level of general knowledge. Similarly, the majority of those with “Mid-level” perceived knowledge (58.3%) had “Mid-level” general knowledge. However, the majority of those with high perceived knowledge (49.5%) were found to have only “Mid-level” general knowledge score. This suggests that participants may have over evaluated their knowledge levels or may have underlying misconceptions in their understanding towards the project. The graphical distribution of GK and PK scores among the participants who have heard of the project is shown in Figure 3.

### 3.3. Trust in Project Wolbachia–Singapore

A total of 77.4% of participants felt that dengue was a serious problem in Singapore. Amongst this group, 59.4% felt that dengue, though serious, is well under control. The majority of participants (64.6%) felt that the Government’s efforts in controlling the local dengue situation are adequate. Public confidence in NEA was also high, with the majority agreeing that NEA acts in the interest of residents (83.2%) and will do its best to help residents (81.6%). Specific to NEA’s role in Project *Wolbachia*, 79.2% of participants expressed that they trust NEA with the project, and 77.4% of the participants claimed they will support the project if NEA recommends the release of *Wolbachia*-*Aedes* mosquitoes (Table 4). The remaining participants either took a neutral stand towards (18.2%, n = 91) or disagreed with (4.4%, n = 22) the release of male *Wolbachia-Aedes* mosquitoes. Among this small minority of 22 participants who objected to the release of *Wolbachia-Aedes* mosquitoes, 15 participants claimed to be aware of Project *Wolbachia* and only 7 participants claimed to know how the *Wolbachia* technology work. The group of 22 respondents who objected to the releases of *Wolbachia-Aedes* mosquitoes also had corresponding low GK scores (average of 2.59 out of 11), which is lower than GK scores among those who did not object (average of 4.44 out of 11). This group of respondents (n = 22) did not feel that their risk of getting mosquito-borne infectious diseases such as dengue and Zika is high.

### 3.4. Acceptance towards Project Wolbachia–Singapore

To further explore the acceptance levels of participants towards the project despite some of the inconveniences they may experience, participants were asked how long they would accept more mosquitoes in their living areas due to Project *Wolbachia*–Singapore. Slightly more than half (56%) of the participants reported that they would be willing to accept the sighting of more mosquitoes for at least half a year for the benefit of reducing dengue risk (Table 5).

### 3.5. Socio-Demographic Factors Associated with Greater Awareness, High Levels of Knowledge, Trust and Acceptance towards the Project

Bivariate and multivariable analysis were used to explore potential associations of socio-demographic factors with the outcomes of interest (Table 6). Each outcome of interest was divided into two groups: awareness (those who indicated being aware of Project *Wolbachia* and those who did not), knowledge (high GK scores versus low and mid GK scores), trust (those who scored 4 and above were deemed as having high levels of trust), and acceptance (those who indicated being able to accept Project *Wolbachia* for at least half a year were deemed as having high levels of acceptance).

Using bivariate analysis, higher education levels (pre-university and above) and longer exposure to the project (>2 years) were associated with higher awareness levels (OR = 1.371, 95% CI = 0.906, 2.075, *p* = 0.135 and OR = 1.826, 95% CI = 1.209, 2.756, *p* = 0.004 respectively). However, multivariable analysis showed that longer exposure to the project was the only independent factor significantly associated with higher awareness levels (adjusted OR = 1.799, 95% CI = 1.190, 2.719, *p* = 0.005).

Higher education levels (pre-university and above) and longer exposure to project (>2 years) were also associated with high levels of general knowledge on the topic of *Wolbachia* technology (GK scores between 8 to 11 points) based on bivariate analysis (OR = 2.994, 95% CI = 1.842, 4.866, *p* < 0.001 and OR = 1.486, 95% CI = 0.929, 2.377, *p* = 0.097 respectively). However, subsequent multivariable analysis showed that higher education levels, was the only independent factor significantly associated with higher knowledge levels (adjusted OR = 2.945, 95% CI = 1.810, 4792, *p* < 0.001).

Respondents willing to accept the sighting of more mosquitoes for at least half a year for the benefit of reducing the risk of dengue are categorised as participants with high levels of acceptance towards the project. Based on bivariate analysis, younger age (age ≤ 40), higher education levels (pre-university and university) and longer of exposure to the project were factors associated with high levels of acceptance towards the project (OR = 0.400, 95% CI = 0.276, 0.579, *p* < 0.001, OR = 0.399, 95% CI = 0.278, 0.575, *p* < 0.001 and OR = 0.771, 95% CI = 0.541, 1.098, *p* = 0.149 respectively). The multivariable analysis showed that younger age and higher education levels were two factors associated with higher acceptance levels (adjusted OR = 0.548, 95% CI = 0.356, 0.843, *p* = 0.006 and adjusted OR = 0.546, 95% CI = 0.357, 0.834, *p* = 0.005 respectively) after considering the effect of all three demographic factors.

No socio-demographic factors were significantly associated with trust in the project.

## 4. Discussion

In a previous household perception survey conducted in study sites, the majority of respondents (69–72%) were aware of Project *Wolbachia* [23]. In this paper, we reported a similarly large majority (74.8%) being aware of the project in the study sites, with greater awareness among respondents with more than two years of exposure to the project compared to less than a year of exposure (80.4% and 69.2% respectively). This result was encouraging as it indicated that the levels of awareness had been maintained over the years even after expanding the site to include more than three times as many households. Longer exposure to the project was found to be independently associated with higher awareness in this study, possibly due to longer periods of intensified publicity campaigns in the study area. High levels of trust (79.2%) and support (77.4%) for the project were also observed in this survey. These results are a testament to the efforts NEA put into community outreach and engagement in Project *Wolbachia–*Singapore.

Whilst the finding that longer exposure to the project was associated with higher awareness levels is intuitive, it is interesting to note that longer exposure was not independently associated with higher levels of knowledge on *Wolbachia-Aedes* technology. Instead, higher education levels (pre-university and above) were found to be independently associated with higher levels of knowledge on *Wolbachia* technology. Respondents with more years of formal education may be more likely to be better-read and hence more informed of current affairs and new technologies. This is consistent with various knowledge, attitude and practice studies conducted on dengue and dengue prevention [28,29,30]. Among respondents who were aware of the project, the majority (76%) were confident of their knowledge in the topic, as revealed by the perceived knowledge (PK) scores. However, the majority of those with high perceived knowledge had only “Mid-level” general knowledge (GK) score, suggesting that participants may overestimate their knowledge in the topic or may have some misconceptions. Weak correlation or lack of correlation between perceived knowledge and actual knowledge is also observed in several studies [31,32,33], where participants overestimate their knowledge in the topic. Only 22.2% of those who were aware of the project obtained a high GK score, suggesting that more communication efforts need to focus on bridging knowledge gap in the population. Based on the questions that were incorrectly answered by majority in this study, the below key messages need to be emphasised in future messaging:*Wolbachia* is a bacterium and *Wolbachia-Aedes* mosquito is not genetically modified*Wolbachia-Aedes* suppression targets only *Aedes aegypti* mosquitoesContinuous releases male of *Wolbachia-Aedes* mosquitoes is required to reduce dengue mosquito population in the long term.

The misconception that the project involves the release of genetically modified mosquitoes could undermine community support, given the public resistance reported towards use of genetically modified mosquitoes for disease control in other countries [34,35]. Furthermore, when the public has the false impression that “*Wolbachia-Aedes* suppression targets many species of mosquitoes”, they may become complacent and relax personal mosquito control efforts. This public complacency had been suggested as a potential risk points in an earlier survey and risk assessment [17,36], although only the minority of the respondents showed signs of complacency. Additionally, such misinformation may lead the public to doubt the project’s success as they may mistakenly expect bites from other species of mosquitoes to be reduced. False impression of *Wolbachia-Aedes* technology as a silver bullet and ignorance of how the technology works could partly explain why participants were unaware that continuous releases were required to sufficiently suppress the *Aedes aegypti* mosquito population in the long term. This has implications on how accepting residents are towards the project, along with the inconvenience they may experience. Therefore, there is a need to address these misconceptions as they may affect the sustainability of the project.

High levels of trust in the project were also reported in this study. This may be due to the general high levels of trust and confidence felt by the public towards Singapore’s Government (a recent study of public confidence in government ranked Singapore second out of 12 countries [37]). In our study, although no socio-demographic factor was significantly associated with high levels of trust in the project, suggesting balanced support among the participants, younger age and higher education levels were associated with higher acceptance levels. Whilst the percentage of supportive participants for the project is considerably high (77.4%), a slightly lower percentage (56%) of the participants were willing to accept the sighting of more mosquitoes for at least half a year for the benefit of reducing dengue risk. We, therefore, inferred that the remaining 44% of the participants may not be as welcoming if the project extends beyond half a year. Some of the feedback received revealed that nuisance caused by sightings of male *Wolbachia-Aedes* mosquitoes and inconveniences such as having to close doors and windows to reduce an influx of mosquitoes, and additional spraying of insecticides and repellent to reduce swarming of male *Wolbachia-Aedes* indoors, could be reasons behind the hesitancy towards having *Wolbachia-Aedes* releases extending beyond half a year in their neighbourhood. To manage community’s expectations, one way is to advocate the project as a scientific project guided by field data on *Aedes aegypti* mosquito population, where possible reduction in mosquito releases is explored when the *Aedes aegypti* mosquito population in the area is low. This may encourage support for the project in long term and motivate residents to proactively carry out steps to prevent mosquito breeding.

Even with high levels of trust and support towards the project, there is a small minority (4.4%) who objected to the release of *Wolbachia*-carrying mosquitoes, and this cannot be overlooked. This group may be aware of the project, but may not be fully informed of the function and benefits of the technology as shown from their low average GK scores. One study showed that knowledge influenced receptivity towards *Wolbachia* technology as a vector control tool trialed in Malaysia [38]. Therefore, while awareness of the project is high, it is also important that the public has accurate and factual knowledge on the technology, to ensure positive advocacy for the project. Noteworthily, all the respondents who objected to the project in this study also felt that their risk of getting mosquito-borne infectious diseases such as dengue and Zika is not high. The low perceived risk of mosquito-borne infectious diseases among these respondents may stem from Singapore’s long-running and largely successful vector control program, which has brought down the *Aedes* house index to a low level of about 2% [8]. A follow up study can be explored to further investigate how perceived risk of mosquito borne disease could influence support for such novel vector control tools.

Other than identifying knowledge gaps to facilitate addressing myths and misconceptions, our findings also pointed out the need for enhanced communications to, firstly, educate the public that the use of *Wolbachia-Aedes* technology is not a silver bullet and that continuous weekly release is essential to bring down the mosquito population and keep the *Aedes aegypti* mosquito population suppressed. Secondly, there is need to share with the public that the number of mosquitoes released may be adjusted when the mosquito population is suppressed to low levels; and lastly, to explore alternative forms of communications to reach out to older and less educated population.

This survey was completed in February 2020, two months prior to a “circuit breaker” (i.e., stringent social distancing measures and cessation of non-essential work activities) in Singapore during the COVID-19 pandemic, and a few months before Singapore saw an unprecedented dengue outbreak. It was fortuitous that the majority (77.0%) of the respondents had heard of Project *Wolbachia* through the public communication which included publicity materials (such as brochures distributed to all households, banners and posters put up at strategic areas in the community) as well as mainstream media, instead of through face-to-face engagement methods. This provided reassurance to step up engagement in the community effectively through these publicity channels during the circuit breaker, when close contact engagement proved difficult. The restrictions on face-to-face contact and the switch to working and studying from home, made it necessary to calibrate the content of the print materials, and also enhance social media engagement, and use alternative modes of communications such as mobile messaging and videoconferencing platforms. The insights from the survey guided our communications and engagement approach but was also contextualised amidst an overwhelming burst of information on COVID-19 and the communications on the dengue outbreak in 2020. It would be interesting to explore a follow up study on how the worst dengue outbreak in 2020 as well as the COVID-19 circuit breaker could have possibly influenced peoples’ perceived risk of mosquito-borne diseases and also their support in the project.

This study has several limitations. Firstly, the surveyors were introduced as representatives from a government authority. Therefore, participants who agreed to participate in the survey may present themselves as being more supportive of the local government and provide more positive responses. Notwithstanding that responses may be systematically biased towards more positive responses, participants who are more resistant towards the project could also use the survey platform to voice their concerns. Secondly, the survey was administered during the day where most of the working population would be at their workplaces away from home, hence suggesting possible bias towards the non-working and older population. Thirdly, the survey was conducted largely in English (the main language of instruction in Singapore) and some residents were unable to participate due to language barriers. However, the survey population (Chinese—64.4%, Malay—18%, Indian—13.2%, Others—4.4%) was largely representative of the general population in Singapore in 2019 (Chinese—76%, Malay—15%, Indian—7.5%, Others—1.5%).

As the project progresses it would be also useful to evaluate the feedback received from the study sites to obtain a deeper understanding of the motivations behind their receptivity or resistance towards the project and factors that shaped attitudes over time. Addressing public concerns through customised communications and proactive engagement remains important as it has implications on the operations and sustainability of the project.

## 5. Conclusions

This study revealed high awareness and support for Project *Wolbachia–*Singapore in the study sites—a strong testament to the communications performed to date amongst the affected communities. Participants with longer exposure to the project (>2 years) and higher education levels (pre-university and above) were found to have better awareness of the project and higher knowledge on *Wolbachia-Aedes* technology respectively. The study also highlighted a knowledge gap among the population, suggesting that communications to address misinformation are needed. As the project continues to expand to cover 15% of total number of public housing apartment blocks in Singapore, and into new towns with high dengue risk, it is essential to conduct future intervention studies to address the knowledge gaps and also evaluate the awareness and support levels in these new areas. It would also be of value to evaluate the perception from the public towards such public health intervention measures especially under the pressure of COVID-19 and large dengue outbreaks.

## Figures and Tables

**Figure 1 ijerph-18-11997-f001:**
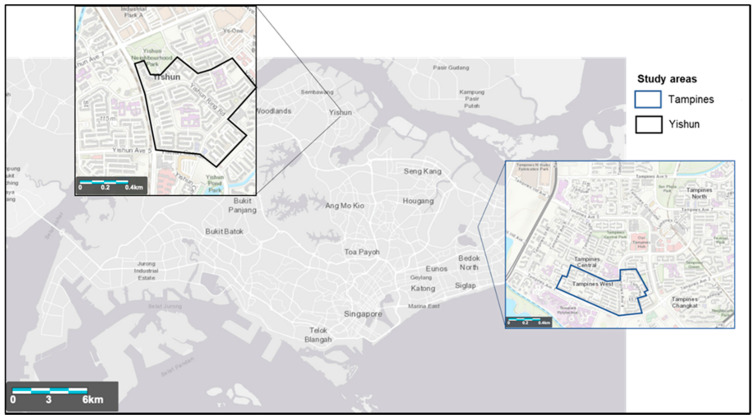
Locality of Project *Wolbachia–*Singapore’s study sites. The base layer was obtained from https://landsatlook.usgs.gov/ (accessed on 22 August 2020).

**Figure 2 ijerph-18-11997-f002:**
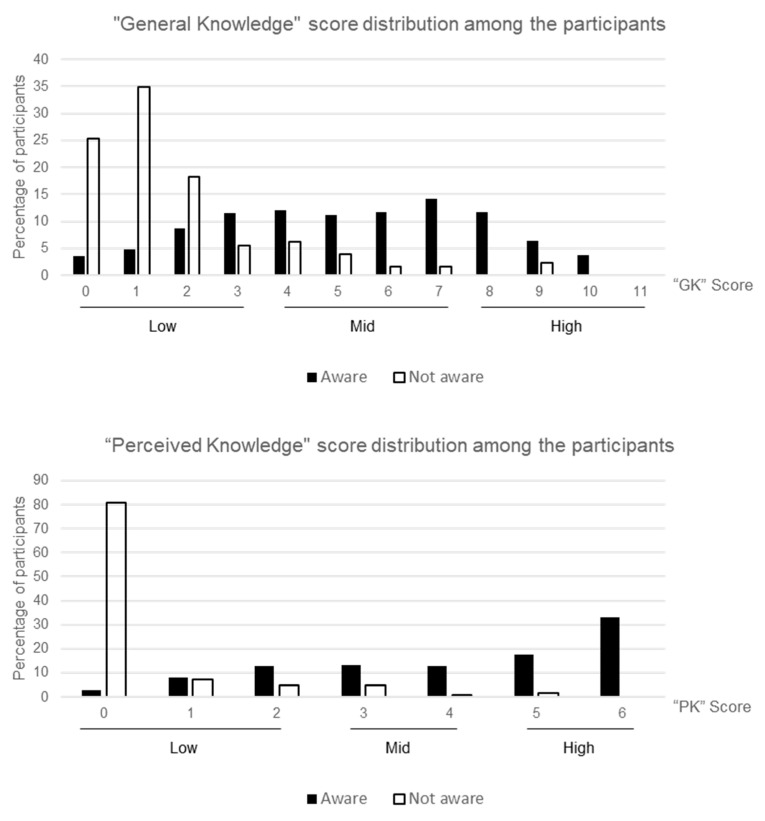
Level of general knowledge (“GK” score) and perceived knowledge (“PK” score) on the *Wolbachia-Aedes* technology among the participants.

**Figure 3 ijerph-18-11997-f003:**
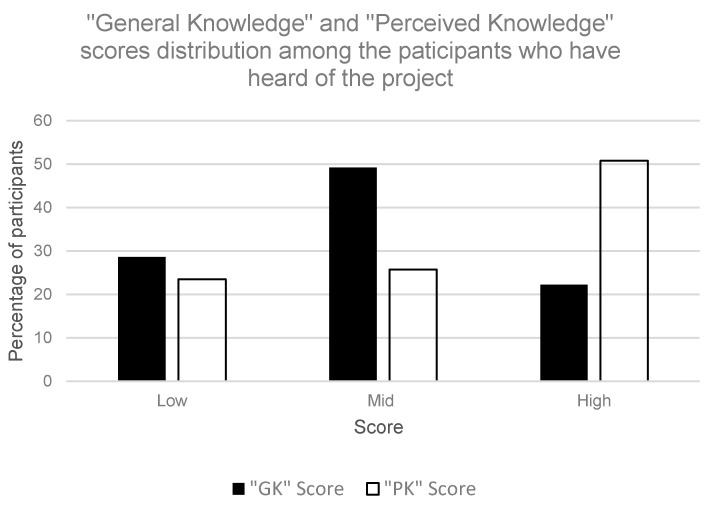
Distribution of general knowledge (GK) and perceived knowledge (PK) scores among the participants who have heard of the project.

**Table 1 ijerph-18-11997-t001:** A total of 500 household surveys were conducted at Tampines and Yishun study sites.

Study Sites	Number of Surveys (Minimum Sample Size)
Tampines	
Site with >2 years of *Wolbachia-Aedes* releases (total no. of households: 2940)	125 (94)
Site with <1 year of *Wolbachia-Aedes* releases (total no. of households: 2620)	100 (93)
Yishun	
Site with >2 years of *Wolbachia-Aedes* releases (total no. of households: 3234)	125 (94)
Site with <1 year of *Wolbachia-Aedes* releases (total no. of households: 4241)	150 (94)
Total	500

**Table 2 ijerph-18-11997-t002:** Socio-demographics of participants (N = 500).

Categories	Participant Distribution, N = 500 (n %)
Gender	
Male	198 (39.6%)
Female	302 (60.4%)
Age (years)	
≤20	35 (7.0%)
21–30	66 (13.2%)
31–40	89 (17.8%)
41–50	77 (15.4%)
51–60	79 (15.8%)
>60	154 (30.8%)
Education Level	
Primary School Leaving Examination (PSLE) and below	123 (24.6%)
‘O’-Levels or equivalent	162 (32.4%)
Pre-university	10 (21.6%)
University	107 (21.4%)
Length of Exposure to Project	
>2 years	250 (50%)
<1 year	250 (50%)

**Table 3 ijerph-18-11997-t003:** Categorised perceived knowledge (PK) and general knowledge (GK) scores among participants who have heard of Project *Wolbachia–*Singapore.

**Perceived Knowledge**	**% of Participants Who Answered “Yes”**			**PK Score**
I know about Project *Wolbachia*–Singapore	91.4%			Low (0–2 points): 23.5% Mid (3–4 points): 25.7% High (5–6 points): 50.8%
I am aware of the reasons why NEA launches Project *Wolbachia*–Singapore	85.0%			
I know the mechanics of how *Wolbachia* technology would work	60.4%			
I know the effects of *Wolbachia* on mosquitoes	61.0%			
I know the consequences of *Wolbachia* on human health	47.6%			
I have read the materials from NEA about Project *Wolbachia*–Singapore	64.7%			
**General Knowledge**	**% of Participants Who Answered Correctly**	**% of Participants Who Answered Incorrectly**	**% of Participants Who Indicated “Do Not Know”**	**GK Score**
*Wolbachia* is a bacterium (True)	30.2%	17.1%	52.7%	Low (0–3 points): 28.6% Mid (4–7 points): 49.2% High (8–11 points): 22.2%
*Wolbachia* is safe (True)	61.0%	7.5%	31.5%	
All mosquitoes regardless to their gender could bite (False)	52.1%	28.6%	19.2%	
*Wolbachia-Aedes* suppression targets many species of mosquitoes (False)	22.2%	32.9%	44.9%	
*Wolbachia-Aedes* mosquito is not genetically modified (True)	21.9%	26.5%	51.6%	
Not all mosquitoes transmit dengue equally (True)	69.5%	8.8%	21.7%	
Mating between *Wolbachia-Aedes* males and wildtype urban females result in eggs that do not hatch (True)	59.6%	4.3%	36.1%	
Project *Wolbachia*–Singapore is being deployed all over Singapore (False)	36.1%	36.1%	27.8%	
Project *Wolbachia*–Singapore involves the release of both male and female *Wolbachia*-carrying *Aedes* mosquitoes (False)	58.0%	13.9%	28.1%	
Male *Wolbachia*-carrying *Aedes* mosquitoes can help reduce dengue mosquito population (True)	73.8%	3.2%	23.0%	
We need to release male *Wolbachia-Aedes* mosquitoes only once to effectively reduce dengue mosquito population in the long term (False)	38.8%	28.1%	33.2%	

**Table 4 ijerph-18-11997-t004:** List of question items under “trust in project” composite measure. All items are measured on a 5-point Likert scale (1 = strongly disagree, 5 = strongly agree).

Question Items	% of Participants Who Strongly Agree or Agree with the Item (N = 500)
I believe that the release of male *Wolbachia*-*Aedes* mosquitoes by NEA has been supported by scientific evidence	72.4
I trust NEA with regard to the Project *Wolbachia*–Singapore	79.2
If NEA recommends the release of male *Wolbachia*-*Aedes* mosquitoes, I will support it.	77.4

**Table 5 ijerph-18-11997-t005:** Length of project trial that is acceptable in participants’ opinion.

Length of Project Trial Which Participants Are Able to Accept	n (%); N = 500
1 month	110 (22.0%)
3 months	110 (22.0%)
half a year	69 (13.8%)
1 year	57 (11.4%)
>1 year	154 (30.8%)

**Table 6 ijerph-18-11997-t006:** Analysis of socio-demographic factors associated with high levels of awareness, knowledge, trust and acceptance towards the project. Each row presents select ORs from bivariate analysis (Pearson’s Chi-square test).

Categories	N	High Awareness of the Project			High General Knowledge of *Wolbachia* Technology (GK Scores between 8–11 Points)			High Trust in Project			High Acceptance towards Project		
		n (%)	OR (95% CI)	*p*-Value	n (%)	OR (95% CI)	*p*-Value	n (%)	OR (95% CI)	*p*-Value	n (%)	OR (95% CI)	*p*-Value
Gender													
Male	198	147 (74.2%)	Reference		37 (18.7%)	Reference		126 (63.6%)	Reference		107 (54.0%)	Reference	
Female	302	227 (75.2%)	1.050 (0.696, 1.585)	0.816	49 (16.2%)	0.843 (0.527, 1.349)	0.476	210 (69.5%)	1.304 (0.839, 1.906)	0.169	173 (57.3%)	1.141 (0.795, 1.636)	0.475
Age (years)													
>40	310	234 (75.5%)	Reference		53 (17.1%)	Reference		214 (69.0%)	Reference		200 (64.5%)	Reference	
≤40	190	140 (73.7%)	0.909 (0.601, 1.376)	0.653	33 (17.4%)	1.019 (0.632, 1.644)	0.938	122 (64.2%)	0.805 (0.549, 1.179)	0.265	80 (42.1%)	0.400 (0.276, 0.579)	<0.001
Education Level													
‘O’-Levels or equivalent and below	285	206 (72.3%)	Reference		30 (10.5%)	Reference		197 (69.1%)	Reference		187 (65.6%)	Reference	
Pre-university and above	215	168 (78.1%)	1.371 (0.906, 2.075)	0.135	56 (26.0%)	2.994 (1.842, 4.866)	<0.001	139 (64.7%)	0.817 (0.561, 1.190)	0.292	93 (43.3%)	0.399 (0.278, 0.575)	<0.001
Length of Exposure to Project													
<1 year	250	173 (69.2%)	Reference		36 (14.4%)	Reference		163 (65.2%)	Reference		148 (59.2%)	Reference	
>2 years	250	201 (80.4%)	1.826 (1.209, 2.756)	0.004	50 (20.0%)	1.486 (0.929, 2.377)	0.097	173 (69.2%)	1.199 (0.825, 1.743)	0.341	132 (52.8%)	0.771 (0.541, 1.098)	0.149

## Data Availability

The datasets used and/or analysed during the current study are available from the corresponding author on reasonable request.

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
