# Peer review of "A Household-Based Survey to Understand Factors Influencing Awareness, Attitudes and Knowledge towards Wolbachia-Aedes Technology"

_ijerph, 2021, doi:10.3390/ijerph182211997_

Round 1

Reviewer 1 Report

This article reports on household surveys administered Jul 2019 to Feb 2020 in two study areas that queried residents about their impressions and knowledge of ongoing mosquito control efforts that involve release of Wolbachia-carrying Aedes aegypti mosquitoes in Singapore. The investigators report on socio-demographic factors that predict perceived and actual knowledge of, as well as resident perceptions of the control program. Resident approval and trust in the release program remains high, but there are some facts about it that are not well understood. These results will be used to inform future efforts to better educate and inform residents of ongoing control programs.

I must admit the sociological aspects of this study are a bit outside my area of expertise and I am not familiar enough with the politics and demographics of Singapore, so I ask the authors humor me asking the following questions

  • How was 500 households decided to be the number of to be surveyed? Something like Table 1 from [23] might be helpful here?
  • How were the two areas chosen compared to the previous study?
  • Is some measure of annual income appropriate to include in the analyses and is the average different between the two areas? Is income well known in Singapore to correlate/predict one of the variables already included?
  • Why weren't languages spoken in the household included in the analyses? Did all the surveyors speak all four official languages, and can you comment on how excluding households with language barriers may have affected the results?
  • How consistent and ongoing are the messaging campaigns? What are some possible explanations for longer exposure to the project being only independent factor significantly associated with higher awareness levels?

It strikes me that a key success of this study was identifying the questions from the survey that reflect information that most needs to be emphasized in future outreach efforts.

Overall a good example of specific and actionable results for future/ongoing outreach efforts. 

Reviewer 2 Report

Releases of male mosquitoes carrying Wolbachia bacteria are being carried out in trial sites around the world as a tool for dengue control. The success of these releases depends in part on high levels of community awareness and support. The authors conducted a survey of people living in areas of Singapore to assess public awareness, knowledge and attitudes towards male Wolbachia mosquito release programs. The authors found strong support overall for Wolbachia releases but identified several areas where community knowledge could be improved (such as the need for ongoing releases and the fact that Wolbachia releases target only a single species). The study represents an important contribution to our knowledge on public attitudes to Wolbachia releases. The paper is well written and the results are carefully interpreted. I have only a few minor suggestions for the authors to consider.

Line 53 – The releases in Innisfail have now been published. Please consider replacing ref 14 with Beebe et al. 2021 PNAS https://www.pnas.org/content/118/41/e2106828118

Page 10, line 18 – “This group of respondents also had corresponding low to mid GK scores (average of 4.1 out of 11)”

How does this compare to the average for the group that did not object?

Page 10, line 20 – “..felt that their risk of getting mosquito-borne infectious diseases such as dengue and Zika is not high”

How was this quantified? Was this all of the people in this category?
